# Neighborhood Environmental Factors and Physical Activity Status among Rural Older Adults in Japan

**DOI:** 10.3390/ijerph18041450

**Published:** 2021-02-04

**Authors:** Kenta Okuyama, Takafumi Abe, Xinjun Li, Yuta Toyama, Kristina Sundquist, Toru Nabika

**Affiliations:** 1Center for Primary Health Care Research, Department of Clinical Sciences Malmö, Lund University, Jan Waldenströms Gata 35, 20502 Malmö, Sweden; xinjun.li@med.lu.se (X.L.); kristina.sundquist@med.lu.se (K.S.); 2Center for Community-Based Healthcare Research and Education (CoHRE), Organization for Research and Academic Information, Shimane University, 223-8 Enya-cho, Izumo-shi, Shimane 693-8501, Japan; t-abe@med.shimane-u.ac.jp (T.A.); yt-ars@med.shimane-u.ac.jp (Y.T.); nabika@med.shimane-u.ac.jp (T.N.); 3Department of Family Medicine and Community Health, Department of Population Health Science and Policy, Icahn School of Medicine at Mount Sinai, 1 Gustave L. Levy Place, New York, NY 10029-5674, USA; 4Department of Functional Pathology, Faculty of Medicine, Shimane University, 89-1 Enya-cho, Izumo-shi, Shimane 693-8501, Japan

**Keywords:** community center, hilliness, neighborhood environment, older adults, physical activity

## Abstract

(1) Background: Although several neighborhood environmental factors have been identified to be associated with older adults’ physical activity, little research has been done in rural areas where the population is aging. This study aimed to investigate neighborhood environmental factors and the longitudinal change of physical activity status among rural older adults in Japan. (2) Methods: The study included 2211 older adults, aged over 60 years, residing in three municipalities in Shimane prefecture and participating at least twice in annual health checkups between 2010 and 2019. Physical activity was identified based on self-report. Hilliness, bus stop density, intersection density, residential density, and distance to a community center were calculated for each subject. Hazard ratios for the incidence of physical inactivity were estimated using Cox proportional hazards models. (3) Results: We found that 994 (45%) of the study subjects became physically inactive during the follow-up. Those living far from a community center had a lower risk of becoming physically inactive compared to those living close to a community center. When the analysis was stratified by residential municipality, this association remained in Ohnan town. Those living in hilly areas had a higher risk of becoming physically inactive in Okinoshima town. (4) Conclusions: The impact of neighborhood environmental factors on older adults’ physical activity status might differ by region possibly due to different terrain and local lifestyles.

## 1. Introduction

Physical activity (PA) is an important modifiable factor that can prevent non-communicable diseases (NCDs), including cardiovascular diseases, cancers, diabetes, and chronic respiratory diseases; these are the four major NCDs that cause more than 36 million deaths, of which 14 million are premature deaths per year globally [1]. Insufficient PA is directly attributed to large healthcare costs, as well as loss of productivity [2,3]. However, the global prevalence of sufficient PA remains low, i.e., approximately 30 percent (%) of people worldwide are insufficiently physically active [4,5]. Given the strong association between PA and major NCDs, a 10% relative reduction in the prevalence of insufficient PA was set as one of the nine primary targets to be attained by 2025 in the global action plan for NCDs [1]. In 2018, the global action plan for PA was built upon that for NCDs as well as existing PA strategies [6], and a 15% relative reduction in the global prevalence of insufficient PA was set as a primary target by 2030, which is aligned with the Sustainable Development Goals (SDGs) from WHO [7]. Reducing inequalities in PA (SDG10) is one of the major goals in the global action plan for PA, and it is to be achieved by ensuring opportunities for PA to women, older adults, and rural and underprivileged residents [7].

In order to achieve this, we need to know the determinants of PA so that evidence-based public health interventions can be initiated [8]. Numerous efforts have been made to identify determinants of PA, starting from individual level (downstream) factors to social, environmental, and policy level (upstream) factors, and approaching them both may be the most effective way to promote PA [7,9]. Ecological models of PA, which explain inter-relations between those downstream and upstream factors, have previously been a common theoretical framework [10]. Despite large research efforts and established conceptual frameworks, translation of research into intervention remains challenging in many settings, and the set target to reduce the prevalence of insufficient PA does not seem feasible to achieve by 2025 [4,9]. One of the reasons may be that determinants of PA are not clear in many populations, especially among those experiencing unequal opportunities of PA, such as older adults and rural residents. 

The world’s population is aging according to estimates from the UN [11]. Healthy aging, which is defined as “the process of developing and maintaining the functional ability that enables wellbeing in older age”, is therefore one focus of WHO’s work concerning aging, and it is an integral part of SDGs [12]. PA for older adults can be beneficial not only for risk reduction of NCDs but also for physical and cognitive functions as well as psychological and mental well-being [13]. Therefore, promoting PA is one of the most important approaches to achieve healthy aging. A number of studies have been conducted to investigate determinants of older adults’ PA. Studies focusing on neighborhood environmental factors (upstream) have been considered to be of particular importance as older adults tend to spend more time within their neighborhoods [14]. In previous studies, several neighborhood environmental factors, e.g., street connectivity, land use mix, and recreational facilities, have been identified as potential determinants [15,16]. However, evidence from rural settings, where people have fewer opportunities for PA, is still scarce [15,16]. Whether environmental and policy interventions, including city design and land use, for urban areas apply to rural areas is unclear [17]. Several studies have found that such factors are not related to PA in rural areas [18,19,20]. While the definition of rural areas is vague and varies by country, typical rural areas have low population density and small settlements. People living in rural areas are more dependent on having a car and have limited accessibility to recreational PA facilities (e.g., parks, gyms), which results in fewer PA opportunities. The prevalence of insufficient PA and obesity tends to be higher in rural areas than in urban areas [21]. In addition, due to the trends of younger people moving into urban settings, rural areas have an accelerated aging rate. Taken together, it is important to identify determinants of PA among older adults in rural settings [22]. This study aimed to investigate associations between several neighborhood environmental factors and PA in older adults residing in aging Japanese rural communities by using a longitudinal study design.

## 2. Materials and Methods

### 2.1. Study Design

This study is a part of the Shimane CoHRE Study conducted by the Center for Community-based Healthcare Research and Education (CoHRE) at Shimane University and three municipalities (Unnan city, Ohnan town, and Okinoshima town) in Shimane Prefecture, Japan. The definitive difference between city and town in Japan is based on population size: city: ≥50,000; town: ≥8000 and ≤49,999. The sizes and proportions of elderly people in each municipality are the following: Unnan: 553 km^2^ (area size), 36,248 people (population size), 38.3% (people ≥ 65 years old); Okinoshima: 243 km^2^, 13,874 people, 39.8% ≥65 years; Ohan: 419 km^2^, 10,374 people, 44.0% ≥65 years [23,24]. The Shimane CoHRE Study aims for early detection and prevention of chronic diseases by conducting surveys on medical information, lifestyle behaviors, and residential characteristics together with an annual health checkup led by the municipal governments. The data used in this study was based on the years between 2010 and 2019.

### 2.2. Study Subjects

In Japan, people who have non-employer-based health insurance, i.e., national health insurance, are eligible for an annual health checkup in their residing municipality. This study included those who participated at least twice in an annual health checkup between 2010 and 2019 and were 60 years of age or older at the first year’s participation (baseline). We excluded those who did not have physical activity levels of at least 60 minutes per day at baseline, which resulted in a total of 2255 study subjects. We further excluded 44 subjects due to missing data (musculoskeletal disorders, *n* = 17; hilliness, *n* = 27). This resulted in 2211 subjects for the final analysis.

### 2.3. Outcome

Physical activity (PA) was based on a two-choice questionnaire item on PA commonly used in standardized health checkups in Japan and recommended by the Ministry of Health, Labor and Welfare. The question was "Is walking or equivalent physical activity performed in your daily life for 60 minutes or more per day?" The status of PA was measured at once for each year at an annual health checkup. This question has been validated in a previous study. The amount of PA corresponding to three metabolic equivalents or more assessed by an accelerometer was significantly higher in subjects who answered “Yes” to the question than those who answered “No” (*p* < 0.001). The sensitivity and specificity of the question were reported to be 63.9% and 61.7%, respectively [25].

### 2.4. Exposure Variables: Neighborhood Environmental Factors

Hilliness, bus stop density, intersection density, residential density, and distance to a community center were measured by geographic information systems (GIS). Land slope was used to assess the hilliness of the neighborhood, which has been reported to be associated with PA, weight change, and diabetes among older adults [26,27,28,29]. Bus stop density, intersection density, and residential density have also been reported to be associated with PA and health among older adults [15,16]. All these measures were calculated within a 1000 m network buffer from the point of residence of each subject based on the actual street network assessed by GIS. The buffer size of 1000 m has previously been found to be an appropriate neighborhood space where people conduct their activities [30]. Mean land slope was computed within the network buffer as degree in angular unit for each resident. The data of mean land slope was created in 2015, stored in 5th mesh data (50 m:50 m grid), and obtained from the National Land Numerical Information administered by the National Land Information Division, National Spatial Planning, and Regional Policy Bureau of Japan. The number of bus stops, intersections with three or more legs, and households were computed within the network buffer and used as a measure for bus stop density, intersection density, and residential density. Distance to a community center was also calculated based on street networks from each point of residence to a specific community center of the subject’s community district. A community center is a place for community residents to participate in a variety of activities such as sports, hobbies, and volunteer activities. Those who are interested in such activities can freely join at a community center within their residential community district. Use of community centers and perceptive accessibility to community centers has been found to be associated with better physical function and walking speed among older Japanese adults [31,32]. The calculated values were categorized into quartiles as follows: Land slope (degree): Quartile 1 (lowest) = 3.41, 6.98; Quartile 2 (low) = 6.98, 10.1; Quartile 3 (high) = 10.1, 13.7; Quartile 4 (highest) = 13.7, 26.2. Bus stop density (count): Quartile 1 (lowest) = 0, 2; Quartile 2 (low) = 2, 4; Quartile 3 (high) = 4, 7; Quartile 4 (highest) = 7, 27. Intersection density (count): Quartile 1 (lowest) = 0, 9; Quartile 2 (low) = 9, 20; Quartile 3 (high) = 20, 40; Quartile 4 (highest) = 40, 126. Residential density (count): Quartile 1 (lowest) = 0; Quartile 2 (low) = 1, 42; Quartile 3 (high) = 42, 166; Quartile 4 (highest) = 166, 1607. Distance to community center (meter): Quartile 1 (closest) = 0.97, 835; Quartile 2 (close) = 835, 1540; Quartile 3 (far) = 1.540, 3.030; Quartile 4 (farthest) = 3030, 11200. Data were created in 2015, stored as point data, and obtained from ArcGIS data collection administered by Esri Corporation, Tokyo, Japan (Esri Japan). All spatial analyses were done using ArcGIS Pro 2.0 (Esri Japan).

### 2.5. Covariates

Age (60–69, 70–79, or 80 and above), sex (male or female), smoking (yes or no), drinking (yes, occasionally, or no), BMI (continuous), musculoskeletal disorders (yes or no), and residential municipality (Unnan, Ohnan, or Okinoshima) were included as covariates. All of this information was collected using self-reported questionnaires, face-to-face interviews, and onsite objective measurements. 

### 2.6. Statistical Analysis

Descriptive statistics were calculated for all variables by PA status at the baseline survey (any of 2010–2019). Person-years were calculated from the start of follow-up until the incidence of outcome (becoming physically inactive assessed by changes in PA status over time), loss of follow-up, or the end of the study period in 2019. Cox proportional hazards model were conducted to estimate the hazard ratio (HR) with 95% confidence intervals (CIs) for the incidence of becoming physically inactive by each neighborhood environmental factor as a primary exposure variable, i.e., land slope, bus stop density, intersection density, residential density, and distance to community center. Cox proportional hazards models are commonly used in public health research to estimate the effect of several variables from the time until a specific outcome (e.g., physical inactivity, functional disability, mortality) takes place [33,34,35,36]. For each neighborhood environmental factor, the correlation with the other environmental factors was verified, and a high correlation was confirmed (r ≥ 0.4, Appendix A). Due to multicollinearity concerns, each factor was analyzed separately in the models. Analyses were performed with models as follows (Table 1):

While other survival analysis methods, such as random survival forest, could also be applied to our data, we chose Cox proportional hazards models in order to be consistent with previous studies investigating other health-related outcomes [34,35,36]. Although, for example, random survival forest may have advantages when handling complex and high-dimensional data [37], it has not yet been confirmed whether it is superior to Cox proportional hazards models for all types of health outcomes, such as cancer or cardiovascular disease incidence [33,38,39]. In this study, we selected five neighborhood environmental variables based on previous neighborhood research and applied Cox models for the ease of consistent interpretation of our results with previous studies [34,35,36,40,41]. Considering different features of the residential municipalities, we performed subgroup analysis by each residential municipality. Finally, considering a natural aging effect of the study subjects, i.e., older adults tend to become more inactive as they age, subgroup analysis by age group (in addition to residential municipality) was conducted (Appendix A). Cut off for the age was set at 75 years as this is the age when the medical insurance system switches to the Medical Insurance System for the Latter-Stage Elderly in Japan, and it was also found to represent a point when elderly people decline in their activity and functional levels [42,43,44,45]. All statistical analyses were conducted using R 3.6.1.

### 2.7. Ethical Considerations

The study protocol was approved by the Ethics Committee of Shimane University (#3912), and written informed consent was obtained from all study subjects prior to the study.

## 3. Results

Table 2 shows subjects’ characteristics at the first time of participation in the survey (baseline). At baseline, all 2211 participants had adequate PA levels. During the observational period, 994 (45.0%) of them became physically inactive.

Appendix A show the Kaplan–Meier curves for the time until becoming physically inactive by quartile category of each environmental factors. In the Kaplan-Meier curves by distance to a community center (Appendix A), higher survival probability was observed in the 4th quartile over time (years).

Table 3 shows the results of the Cox proportional hazards model examining the associations between neighborhood environmental factors and becoming physically inactive, i.e., incidence of physical inactivity. After adjusting for age, sex, smoking, drinking, and BMI (Model 2), the risk of becoming physically inactive was significantly higher in the 2nd quartile (HR = 1.16; 95%CI = 1.00, 1.33) and the 3rd quartile (HR = 1.18; 95%CI = 1.01, 1.36) compared to the 1st quartile for the variable bus stop density, i.e., the risk of becoming physically inactive was higher among those living in areas with high bus stop density compared with those living in areas with low bus stop density (Model 2). For distance to a community center, the risk of becoming physically inactive was significantly lower in the 4th quartile (HR = 0.75; 95%CI = 0.57, 0.93) compared to the 1st quartile. This means that living farthest away from a community center was associated with a significantly lower risk of becoming physically inactive than living closer to a community center (Model 2). After adjusting for all covariates (Model 3), incidence of physical inactivity was no longer significantly associated with any neighborhood environmental factor.

Figure 1 and Figure 2 show the results of subgroup analyses by the three residential municipalities. Compared with a shorter distance (1st quartile), a longer distance (4th quartile) from home to the community center was associated with a lower incidence of physical inactivity in Ohnan town (HR = 0.66; 95%CI = 0.47, 0.93) (Figure 1). In Okinoshima town, there was an increased risk of physical inactivity from living in hilly areas as observed in the 2nd (HR = 1.55; 95%CI = 1.01, 2.4) and 3rd quartile (HR = 1.66; 95%CI = 1.08, 2.55) compared to living in non-hilly areas (1st quartile) (Figure 2).

Appendix A shows the results of the sensitivity analysis further stratified by age group. Results were mostly consistent to the ones without age stratification, except for the 4th quartile of land slope among those over 75 years in Okinoshima town: HR = 3.13 (95%CI = 2.09, 4.19).

## 4. Discussion

We found that bus stop density and distance to a community center were significantly associated with the risk of becoming physically inactive, but the significance disappeared after adjustments in the final analyses. By analyzing this further via residential municipality, we found that these associations varied by municipality, and another environmental factor—hilliness—was associated with risk of physical inactivity.

The association between bus stop density and physical inactivity was inconsistent with previous studies. Bus stop density can be used as a measure of public transportation access, and a systematic review found that better access (high density) was associated with higher PA levels [16]. Public transportation could be regarded as partial active transportation as it engages walking or cycling from one’s home to, for example, a bus stop/station, and from a stop/station to a destination [46,47]. Few studies have examined public transportation access and PA in rural settings, where public transportation is often not a primary mode of transportation for the residents. One study found that those who live in an area with better public transportation access have higher PA levels in rural settings [48]. On the other hand, one study found that drivers had higher PA levels than non-drivers among rural older adults [49]. It may be possible that having a car is important for older adults in rural areas as it helps them to reach destinations where they can engage in PA, such as gyms and parks. However, altering the mode of transportation from driving to walking, cycling, and public transportation is still important as it can benefit people’s chronic health as well as the environment by reducing CO_2_ emissions. In addition, driving status was also measured as an effect modifier in the previous study, and public transportation access was especially beneficial for non-drivers among rural older adults [48]. It remains unclear, based on results from our study, whether such a common mechanism exists, as we were not able to account for driving status. However, it is useful to know that the association may not be similar in urban and rural settings, which may help to plan appropriate environmental interventions.

The results of this study also demonstrated that living close to a community center may increase the risk of physical inactivity. In Japan, a community center functions as a place where people gather for social interactions, volunteer activities, and local events, and the role of these centers is particularly important in rural areas. Previous studies have reported that social participation and interaction with people are important for maintaining and promoting PA [50]. In addition, physical proximity to such places was associated with social participation and interaction [51]. However, our findings were in contrast to our hypothesis and, although we cannot entirely explain why this is the case, some explanations may exist. First, social interactions in rural areas, if too excessive, can have negative effects on health and behaviors [52]. Previous studies have reported that an experienced obligation to participate in community events could lead to a psychological burden and have a negative effect on physical and mental health. Second, there are smaller community units than the community center districts. In the areas far from the community center, it is possible that the social activities in those smaller units are more pronounced in order to compensate for the lack of access to a community center. As such activities within those smaller community units are often autonomous and occur spontaneously, we could not capture them in this study. In future studies, it may be necessary to consider activities in smaller units as well. Third, residents in areas far from community centers may be engaged in other types of PA that could not be determined by questionnaires. Living far from a community center also indicates living far from a city center, and such residents may be more likely to engage in farming. According to the American College of Sports Medicine, farming is one of the vigorous PA activities (≥6 METs) [53]. Although it is unknown whether the respondents included farming as PA in their answers in this study, it is necessary to account for it in future studies. Fourth, it is possible that residents living far from a community center drive cars and that this helps them to reach destinations where they can conduct PA.

In stratified analysis by residential municipality, the effect of bus stop density on physical inactivity disappeared, but that of distance to community center remained in Ohnan town. It is worthwhile notifying that the effect is different by municipality and that different municipalities often have different features in terms of demographics, geography, and culture. For example, the proportion of forestry (non-residential) area in Unnan is 79.1%, in Okinoshima, 86.6%, and in Ohnan, 86.4% [24]. While it is difficult to make assumptions from these general statistics, it may be important to account for differences in geographic features in more detail.

In Okinoshima town, where fishing is the main industry, the residents who lived in hilly areas had higher risk of physical inactivity. This result is inconsistent with a previous cross-sectional study that found that living in a hilly neighborhood was positively associated with PA among older Japanese adults [26]. The positive association between hilliness and PA was hypothesized to be due to beautiful scenery in a hilly environment encouraging older people to conduct recreational PA. In a previous study, hiking trails in hilly environments were reported to encourage PA among the residents of rural areas [54]. On the other hand, it was reported that uphill walking has a higher perceived exertion and physical load [55]. In addition, hilliness has been reported as a risk factor for weight gain, and it has been associated with lower recreational PA (sports) among older adults [28,29]. It is therefore inconclusive whether hilliness could be a barrier or promoting factor for PA and health as the perception of hilliness might differ between individuals. Older people who have lived and worked in a hilly environment during their life may be less likely to perceive such terrain as a barrier [56]. Furthermore, given that some previous studies have found that hilliness is a health-promoting factor, i.e., hilly neighborhoods had a preventing effect for type 2 diabetes, it is important to investigate its effect by assessing whether there is a clinically meaningful cut off for the degree of hilliness (slope) [27].

In our study settings, hilliness was associated with PA only in Okinoshima, where fishing is more common than farming compared to the other municipalities (Unnan: 7820 t (amount of yield from rice), 0 t (fishing); Ohnan: 4950 t (amount of yield from rice), 0 t (fishing); Okinoshima: 1410 t (amount of yield from rice), 52,215 t (fishing)). This could reflect that hilliness may be a barrier for being physically active for those who do not engage in farming, which is more often observed in hilly and isolated areas. However, our study was insufficient for such conclusions as we did not measure perceptions of the environment as well as local lifestyles, e.g., whether residents engaged in farming, and other characteristics. The effects of different environmental factors depending on local lifestyles and characteristics are particularly important for the generalization of environmental intervention policies [14] and need to be considered in more detail by future studies.

This study has several strengths. First, this study is one of the very few studies that longitudinally assessed the association between several neighborhood environmental factors and physical activity among older adults in rural areas. Second, this study used objective measures of each neighborhood environmental factor by applying GIS. In addition, each factor was measured based on pinpoint residential address as well as precise street network data. In order to improve policy implications, investigating mediators as well as comparing effects of other neighborhood environmental factors would be warranted for future studies [57,58]. Third, the study was conducted within highly aged populations, which is important for future policies. The findings of this study would thus be a useful basis for future investigations in other settings.

This study also has several limitations. First, we could only use data concerning environmental factors at one time point. It is possible that older adults with better health may choose to live in more comfortable areas over time, or vice versa. Some environmental factors, such as bus stop density and intersection density, might change over time as well, and the study design may need to be more refined and take neighborhood changes and people’s mobility into account in the future. Second, sampling took place in multiple centers and recruited study participants from an annual health checkup; the latter may have resulted in selection bias. Those who participate in annual health checkups may be more conscious of their health (which does not necessarily mean healthier) compared to those who do not participate in checkups. As a result, the absolute risk of becoming physically inactive might have been underestimated. The lack of data to identify reasons regarding failure to follow-up might have missed several subjects who became physically inactive. Third, we assessed PA using self-report questionnaires, which may have over- or under-estimated PA as a result of self-report bias. In addition, even though the question was validated in previous studies, more detailed and standardized methods of assessments are needed. For example, the association between neighborhood environmental factors and PA might vary by type of physical activity. It is also important to assess its association with domain-specific PA in future studies. The current study did not assess whether individuals who became physically inactive during the study period increased their activity level later on [59]. This should be further assessed with additional follow-up measurement after the event of physical inactivity occurred. Fourth, we could not account for potential effects of unmeasured factors that may have influenced the association between neighborhood environment and PA, e.g., subjective measures of other environmental factors and socioeconomic status. Fifth, it is possible that some of our statistically significant findings could have occurred by chance. Finally, while we measured five neighborhood environmental factors which were hypothesized to be important for older adults’ PA based on previous research, it is necessary to collect a larger variety of environmental factors to conduct more complex analysis in order to identify previously unknown environmental factors [60].

## 5. Conclusions

This longitudinal study found that bus stop density, distance to a community center, and a hilly environment were related to the risk of physical inactivity among rural older adults in Japan. However, the association between the neighborhood environmental factors and physical inactivity differed by residential municipality, which may be due to differences in demographics and geography as well as local lifestyles.

## Figures and Tables

**Figure 1 ijerph-18-01450-f001:**
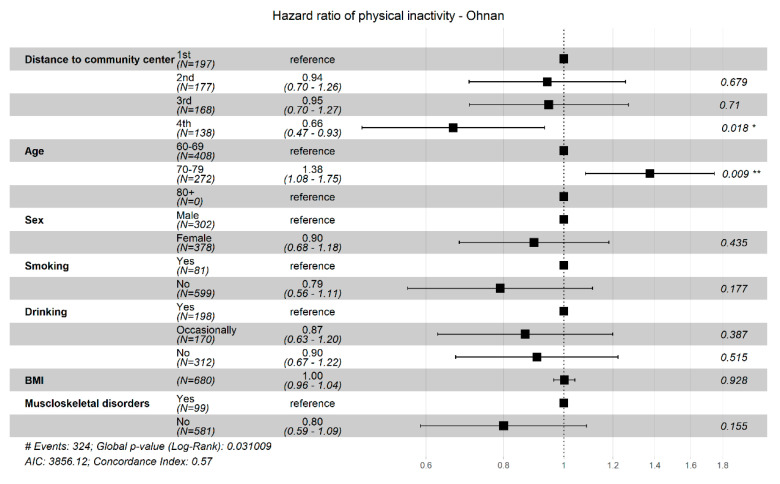
Hazard ratio of physical inactivity by quartile category of distance to community center in Ohnan. Quartile 1 (1st) = 0.97, 835 meter (m); Quartile 2 (2nd) = 835, 1540 m; Quartile 3 (3rd) = 1.540, 3.030 m; Quartile 4 (4th) = 3030, 11,200 m.

**Figure 2 ijerph-18-01450-f002:**
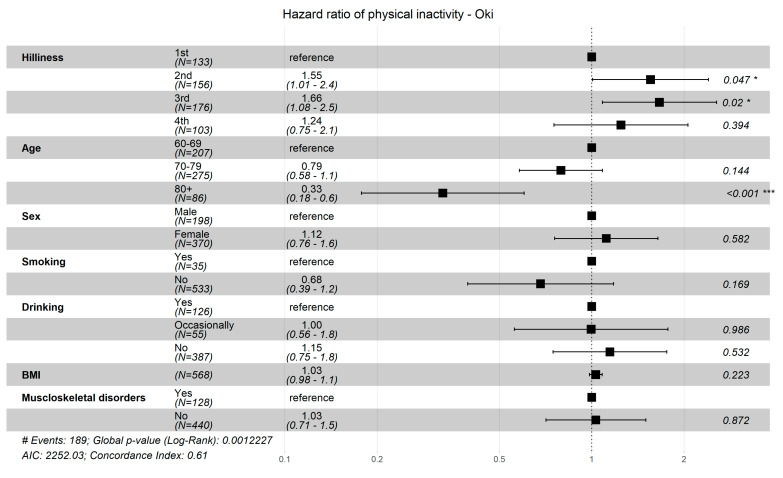
Hazard ratio of physical inactivity by quartile category of hilliness in Okinoshima. Quartile 1 (1st) = 3.41, 6.98 degree(d); Quartile 2 (2nd) = 6.98, 10.1 d; Quartile 3 (3rd) = 10.1, 13.7 d; Quartile 4 (4th) = 13.7, 26.2 d.

**Table 1 ijerph-18-01450-t001:** Analysis models.

	Covariates Adjusted
**Model 1**	Age + Sex
**Model 2**	Age + Sex + Smoking + Drinking + BMI
**Model 3**	Age + Sex + Smoking + Drinking + BMI + Musculoskeletal disorders + Residential density

Outcome: Physical inactivity. Exposure: neighborhood environmental factors.

**Table 2 ijerph-18-01450-t002:** Baseline characteristics of study subjects.

Variables	Became Physically Inactive	
	No (%)	Yes (%)
**N**	1217	994
**Sex**		
Male	520 (42.7)	424 (42.7)
Female	697 (57.3)	570 (57.3)
**Age**		
60–69	538 (44.2)	521 (52.4)
70–79	568 (46.7)	434 (43.7)
80+	111 (9.1)	39 (3.9)
**Municipality of residence**		
Unnan	482 (39.6)	481 (48.4)
Oki	379 (31.1)	189 (19.0)
Onan	356 (29.3)	324 (32.6)
**Smoking**		
Yes	86 (7.1)	88 (8.9)
No	1131 (92.9)	906 (91.1)
**Drinking**		
Yes	320 (26.3)	278 (28.0)
Occasionally	238 (19.6)	213 (21.4)
No	659 (54.1)	503 (50.6)
BMI (mean (SD))	22.4 (2.8)	22.4 (3.0)
**Musculoskeletal disorders**		
Yes	213 (17.5)	164 (16.5)
No	1004 (82.5)	830 (83.5)

**Table 3 ijerph-18-01450-t003:** Cox proportional hazard model for physical inactivity by neighborhood environmental factors.

	Model 1 ^a^	Model 2 ^b^	Model 3 ^c^
	HR (95%CI)	HR (95%CI)	HR (95%CI)
**Slope (ref. 1st, lowest)**			
2nd	1.12 (0.95, 1.30)	1.12 (0.94, 1.29)	1.14 (0.97, 1.32)
3rd	1.10 (0.93, 1.28)	1.10 (0.92, 1.28)	1.13 (0.95, 1.30)
4th (highest)	1.00 (0.82, 1.17)	0.99 (0.81, 1.17)	0.98 (0.80, 1.16)
**Bus stop density (ref. 1st, lowest)**			
2nd	1.15 (0.99, 1.32)	1.16 (1.00, 1.33)	1.09 (0.92, 1.26)
3rd	1.18 (1.01, 1.36)	1.18 (1.01, 1.36)	1.14 (0.96, 1.31)
4th (highest)	1.11 (0.93, 1.29)	1.12 (0.95, 1.30)	1.04 (0.86, 1.22)
**Intersection density (ref. 1st, lowest)**			
2nd	0.94 (0.76, 1.12)	0.93 (0.75, 1.11)	0.94 (0.76, 1.13)
3rd	1.06 (0.89, 1.23)	1.06 (0.89, 1.23)	1.03 (0.86, 1.20)
4th (highest)	1.08 (0.91, 1.26)	1.09 (0.92, 1.26)	1.07 (0.90, 1.25)
**Residential density (ref. 1st, lowest)**			
2nd	1.06 (0.90, 1.23)	1.06 (0.90, 1.23)	1.07 (0.90, 1.24)
3rd	1.07 (0.90, 1.24)	1.06 (0.89, 1.23)	1.07 (0.90, 1.24)
4th (highest)	0.96 (0.78, 1.13)	0.96 (0.78, 1.13)	0.95 (0.78, 1.13)
**Distance to community center (ref. 1st, lowest)**			
2nd	0.94 (0.77, 1.11)	0.93 (0.76, 1.10)	0.96 (0.79, 1.13)
3rd	0.90 (0.73, 1.08)	0.90 (0.73, 1.07)	0.90 (0.73, 1.08)
4th (highest)	0.75 (0.57, 0.93)	0.75 (0.57, 0.93)	0.82 (0.64, 1.01)

Note: ^a^ Model 1: adjusted for age and sex; ^b^ Model 2: adjusted for age, sex, smoking, drinking, and BMI. ^c^ Model 3: adjusted for age, sex, smoking drinking, BMI, musculoskeletal disorders, and city of residence.

## Data Availability

Data cannot be shared publicly because of ethical issues. Data will be available from the CoHRE under permission of the Ethics Committee of Shimane University. Anyone who wants to access the data should contact with the corresponding author.

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
