# Peer review of "Neighborhood Environmental Factors and Physical Activity Status among Rural Older Adults in Japan"

_ijerph, 2021, doi:10.3390/ijerph18041450_

Round 1

Reviewer 1 Report

Overall this paper contributes to the understanding of how neighborhood environmental factors affecting physical activities of senior populaition. My comments are as follows:

  1. The authors should enhance the literature section especially on how survival analysis are used in similar public health studies.
  2. The authors should elaborate the Cox proportional model for readers to better understand the workflow. And a comparison between Cox proportinal model and other survival analysis models like survival random forest should be conducted.
  3. The study subjects are individual senior people. For this particular age group, as they become "older", they tend to be more inactive physically in nature. The paper should consider this factor.
  4. The paper needs thorough English editing.

Reviewer 2 Report

This paper tried explain the dynamics of physical activities of senior population as affected by neighborhood environmental variables. My comments are as follows:

  1. the author should elaborate the Cox proportional hazards model. Briefly distinguishing exposed variables and covariates and the explanation of the model function would help readers understand. And why choose Cox model? There are other models in survival analysis like survival random forest model that might fit better this study.
  2. There is a natural aging effect of the study subjects (the individuals surveyed in the study). As they become older, they "tend" to become more "inactive", the study does not consider this effect/variable.
  3. The manuscript needs thorough English editing.

Round 2

Reviewer 2 Report

Thank you for the opportunity to review this manuscript which aimed to examine associations between neighborhood environmental factors and PA among older Japanese adults residing in rural communities.

Methods:

Line 176. Physical activity is a dynamic behavior that changes over time - thus I question whether Cox proportional hazard ratio is the best analytic approach. Please provide additional information on the frequency of measurement for physical activity, the number of measurements on average per participant, etc. Also, it is unclear if physical activity measures were included after an individual reported physical inactivity. What about those individuals that may have declined but then increased activity again? 

Results:

Lines 205-256: The results of the 4th quartile are described as being significant in the text but not in the table. If the 4th Q is not statistically significant it is misleading to present it so in the text. Also see lines 307-309. 

Lines 256: Please specify what model you are referring to for this interpretation. Again, if finding are not significant the text should be modified accordingly. 

Line 149: Regarding results corresponding to Table 2, the authors report only significantly findings. Please also note that the findings of Model 3 (fully adjusted model) reveal no significant association between becoming physically inactivity and neighborhood environment variables. 

Line 269 (Figures): Please elaborate on why Kaplan-Meir curves are presented for distance to community center but no other environmental variables (e.g., bus stop density). 

Suggested minor edits/comments: 

If space permits, the information presented in limes 96-98 would be better presented in table format. Same for lines 156-162

Limitations: please explore how selection bias might have impacted your findings; specifically, by using those that participate in annual health checkups (compared to those that do not); and those that were already inactive at baseline.  While the measure of PA is reported to be reliable/valid, it is a blunt edge that does not allow for the magnitude of change to be assessed.  

Author Response

Response to Reviewer 2 Comments

Thank you for the opportunity to review this manuscript which aimed to examine associations between neighborhood environmental factors and PA among older Japanese adults residing in rural communities.

Methods:

Line 176. Physical activity is a dynamic behavior that changes over time - thus I question whether Cox proportional hazard ratio is the best analytic approach. Please provide additional information on the frequency of measurement for physical activity, the number of measurements on average per participant, etc. Also, it is unclear if physical activity measures were included after an individual reported physical inactivity. What about those individuals that may have declined but then increased activity again? 

Response: Thank you for these comments. We prefer to keep the Cox regression models (see below, the physical inactivity is counted as an event/incident) but have instead elaborated on our statements further based on your and the Editor’s suggestions. The status of physical activity was measured once per year, i.e., at the annual health checkup. We have now provided this information in the methods section (2.3. Outcome). Regarding those individuals who may have declined but then increased their activity again, we have not assessed this in the study as we set physical inactivity as an event that censored the individual and thus further follow-up of that individual. We have discussed this as a potential limitation and that future studies could continue the follow-up also after physical inactivity has occurred.

Results:

Lines 205-256: The results of the 4th quartile are described as being significant in the text but not in the table. If the 4th Q is not statistically significant it is misleading to present it so in the text. Also see lines 307-309. 

Response: Thank you for noticing! We have now removed this text so that all text is congruent with the information in the table.

Lines 256: Please specify what model you are referring to for this interpretation. Again, if finding are not significant the text should be modified accordingly. 

Response: We have now specified which model is referred to in the sentence. In addition, we have modified the text accordingly.

Line 149: Regarding results corresponding to Table 2, the authors report only significantly findings. Please also note that the findings of Model 3 (fully adjusted model) reveal no significant association between becoming physically inactivity and neighborhood environment variables. 

Response: Thank you for these comments; we have now added the sentence “After adjusting for all covariates (Model 3), incidence of physical inactivity was not associated with any neighborhood environmental factor”. We have also re-ordered the results, i.e. we put the Kaplan-Meier curves first, and then the section for table 2 in order to improve clarity.

Line 269 (Figures): Please elaborate on why Kaplan-Meir curves are presented for distance to community center but no other environmental variables (e.g., bus stop density). 

Response: We have now provided Kaplan-Meier curves for all other environmental variables (Figure S1-S5).

Suggested minor edits/comments: 

If space permits, the information presented in limes 96-98 would be better presented in table format. Same for lines 156-162

Response: We have now provided a table that contains the information in lines 155-163.

Limitations: please explore how selection bias might have impacted your findings; specifically, by using those that participate in annual health checkups (compared to those that do not); and those that were already inactive at baseline.  While the measure of PA is reported to be reliable/valid, it is a blunt edge that does not allow for the magnitude of change to be assessed.  

Response: We appreciate this comment. We have now discussed as a limitation that those who participate in annual health checkups are relatively more health conscious (this does not necessarily mean healthier) compared to those who do not participate in health checkups. As a result, the absolute risk of the outcome (becoming physically inactive) might have been underestimated.